# Prevalence of overweight and obesity among Kuwaiti adolescents and the perception of body weight by parents or friends

**Ahmad R. Al-Haifi[1], Balqees A. Al-Awadhi[1], Yousef A. Al-Dashti[1], Badriyah H. Aljazzaf[1], Ahmad R. Allafi[2], Mariam A. Al-Mannai[3], Hazzaa M. Al-Hazzaa[4]***

**1** Department of Food and Nutrition Science, College of Health Sciences, PAAET, Showaikh, Kuwait,
**2** Department of Food Science and Nutrition, College of Life Sciences, University of Kuwait, Kuwait, Kuwait,
**3** Department of Mathematics, University of Bahrain, Sakheer, Bahrain, **4** Lifestyle and Health Research Center, Health Sciences Research Center, Princess Nourah Bint Abdulrahman University, Riyadh, Saudi Arabia

* halhazzaa@hotmail.com

**Data Availability Statement:** All relevant data are within the paper and its Supporting Information

## Abstract

### Objective

Recently, the State of Kuwait has witnessed a steady rise in the prevalence of obesity among children and adolescents. The present study aims to provide an update on the rate of overweight or obesity among Kuwaiti adolescents and examines the associations between adolescents' overweight/obesity levels and their perception of body weight as seen by parents or friends.

### Methods

A cross-sectional study was conducted in Kuwaiti secondary schools and included adolescents between the ages of 15 and 18 years, using a multistage stratified random sampling method. Body weight and height were measured. A specifically designed self-report questionnaire was used to assess parents' and friends' perceptions of an adolescent's body weight.

### Results

A total of 706 adolescents were included the study. The prevalence of overweight or obesity among Kuwaiti adolescents reached nearly 50%, with males (54.3%) having a significantly higher overweight or obesity percentage than females (44.6%). No significant difference in the prevalence of obesity relative to age, from 15 to 18 years, was found. In addition, logistic regression analysis, adjusted for age and gender, revealed that adolescents perceived their parents (p = 0.011 and p < 0.001) or friends (p = 0.002 and p < 0.001) as more likely to classify their weight as overweight or obese, respectively.

### Conclusion

Overweight or obesity levels appear to be high among Kuwaiti adolescents, and appears to have reached a plateau recently. Efforts to combat obesity and promote physical activity

files. Any additional information, will be provided upon reasonable request.

**Funding:** For your information the corresponding author, Professor Hazzaa M. Al-Hazzaa's research has been funded by the Deanship of Scientific Research at Princess Nourah Bint Abdulrahman University through the Fast-track Research Funding Program.

**Competing interests:** The authors have declared that no competing interests exist.

and healthy nutrition are needed. Future studies should seek to identify important moderators of parental and social underestimation/overestimation of children's overweight or obesity.

## Introduction

Over the past 30 years, the prevalence of overweight and obesity among adolescents worldwide has increased rapidly [1]. It is estimated that about 18% of children and adolescents aged 5–19 were overweight or obese in 2016 [1]. In the State of Kuwait, the prevalence of overweight or obesity was shown to be higher than what was reported elsewhere in the world. For example, a study conducted by Al-Haifi et al. revealed that the prevalence of overweight and obesity was 50.5% in boys and 46.5% in girls [2]. A more recent study indicated that more than half of Kuwaiti adolescents (59.8% of boys, and 49.2% of girls) were overweight or obese [3]. These figures were much higher than the prevalence rate of overweight or obesity reported for boys (30.0%) and girls (31.8%) in 2004 [4]. It is well recognized that obesity in adolescence is associated with numerous immediate and long-term adverse health conditions [5], and that obesity is very likely to persist into adulthood [6]. Adult obesity and associated comorbidities, such as diabetes and cardiovascular disease, currently represent the main costs acquired by health care organizations [7].

Adolescence is a distinct developmental and transition period into adulthood, with increasingly expanding autonomy [8]. Throughout this developmental period, parents and the family continue to have an important and central influence on adolescents' health and lifestyle. However, a study involving American adolescents aged 14–18, with a body mass index (BMI) greater than the 85th percentile for their age and sex, indicated that parents overestimated how supportive they were compared to adolescents' perception [9]. Further, as the social learning theory suggests, the adolescents' decision to embark on a particular behavior depends on the exposure to norms, values and behavioral attitudes of other people with whom they interact [10]. The period from childhood to adolescence is also a time when peers potentially exert a stronger influence on behavior, including the effect on obesity risk behavior [11]. An Australian study examined how friendship network characteristics can be associated with obesity-related behaviors in students aged 11–13. The regression model showed that friendship networks varied by gender and behavior type, including the number of friends positively associated with physical activity intensity (in males) and screen time (in females) [12]. Thus, increased physical activity in different contexts may come from the selection of active male friends, and for girls, having more friends is associated with increased screen time [12].

Compared with peers with normal weight, a greater percentage of adolescents with severe obesity reported lower self-esteem and body satisfaction, parental encouragement to diet, and peer weight teasing [13]. These factors may diminish their quality of life and may predict increased weight gain over time [13]. Moreover, overweight and obese adolescents are more likely to be teased about their weight than average-weight individuals [14–16]. Weight-related teasing is particularly associated with clinical eating disorders, unhealthy weight control behaviors, and negative weight-related attitudes such as body dissatisfaction [17]. Although family members and friends play a role in adolescent body satisfaction and weight control behaviors, they have also been determined to be the two main sources of weight teasing [14, 18–21]. Understanding the associations between an adolescent's overweight or obesity and their perception about how parents or friend see their actual weight status is extremely important because lifestyle behavior intervention programs benefit from parental and social support.

Therefore, the present study reports on the prevalence of overweight or obesity among Kuwaiti adolescents and examines its relationship with their perception of body weight as seen by parents or friends.

## Materials and methods

### Study design and participants

A cross-sectional study was conducted in Kuwaiti secondary schools during the fall of 2019. Recruited adolescents were between the ages of 15 and 18 years. A predetermined sample size of 368 participants for each sex group was calculated, so that the population proportion was assumed to be at 0.60 with a tolerated error of 0.05. A multistage stratified random sampling method was used to select the participants from the schools. The stratification was based on gender (male versus female schools), type of school (public versus private), and geographical or administrative location (governorates) in the State of Kuwait. Two schools (one for boys and one for girls) from each governorate were selected, along with two private schools (sub-total: 14 schools). Then, one classroom from each grade (i.e., 10th, 11th, and 12th) was randomly chosen in each selected school. Inclusion criteria included healthy students not suffering from diet-related disorders (using questions inserted in the questionnaire). Ethical approval was obtained from the Institutional Review Board (IRB) at the College of Health Sciences in Kuwait. In addition, the Ministry of Education in Kuwait approved the study. Written informed ascents were obtained from all participating students and written consents were attained from their parents if they were younger than 18 year-olds. The research procedures were conducted in accordance with the principles expressed in the Declaration of Helsinki.

### Anthropometric measurement

Bodyweight was measured to the nearest 100 gm (Seca 875 Weight Scale, Germany) and height to the nearest 0.1 cm (Seca 213 Standiometer, Germany) by trained researchers assisted by students. All measurements were conducted with minimal clothing and without shoes. BMI was computed as the ratio of weight in kilograms divided by the squared height in meters. The extended International Obesity Task Force (IOTF) age- and sex-specific BMI cutoff reference standards were used to classify underweight, normal weight, and overweight or obesity relative to the adolescent's age [22]. For the analysis, the adolescents were categorized according to BMI classifications: non-overweight/non-obesity versus overweight/obesity. In addition, students were asked to report their age where puberty had occurred to them.

### Description of questionnaire

A specifically designed and previously tested self-report questionnaire was used for this study [23]. The questionnaire consisted of three sections: 1) frequency of media use (e.g., magazines, TV, and internet); 2) influence of media on dieting to lose weight; and 3) perception of body weight by parents or friends. The question in the first section was "How often do you read magazines per week, watch TV per day, and use the internet per day?" The second section aimed to measure the associations between media and dieting with questions such as "Does reading magazines, watching TV, and using the internet influence your diet to lose weight?" The responses included the following choices: high influence, moderate influence, weak influence, or no influence. The third section measured the perception of one's body weight by one's parents or friends through questions such as "How do your parents or friends perceive the adolescent's body weight as underweight, normal weight, overweight, or obese?" Also, participants were asked whether any of their parents or friends teased them because of their weight.

The responses included the following choices: always, sometimes, rarely, or no. In the present study, we used the data from part three only, which was related to adolescent's body weight perception as seen by parents or friends.

## Statistical analysis

Data were entered into an SPSS data file, checked, cleaned, and analyzed using IBM-SPSS program, version 22 (Chicago, IL, USA). Descriptive statistics were obtained for all variables and reported as means and standard deviations or percentages. Differences between males and females in anthropometric measurements were tested using a t-test for independent samples. Chi-square test of independence was used to examine the relationships of selected variables related to adolescents' perception of how parents' or friends' see their weight status relative to an adolescent's overweight or obesity versus non-overweight/non-obesity status. Two-way ANOVA (gender by age category) was used to test differences in BMI across age and gender. In addition, multivariable analyses (MANCOVA) were used to test differences in selected variables stratified by gender and obesity status while controlling for age. Finally, logistic regression analyses with adjusted odds ratio were used to test the differences in selected variables relative to non-overweight or non-obesity versus overweight or obesity, while adjusting for age and gender. The alpha level was set at 0.05, and any value less than 0.05 was considered significant.

## Results

Table 1 describes the anthropometric characteristics of the participants relative to gender. A total sample of 343 boys and 363 girls was obtained for the study. There were significant differences between males and females in body weight ($p < 0.001$), height ($p < 0.001$), and BMI ($p = 0.029$). Only 7% of the adolescents were underweight, whereas 25.4% were obese. Moreover, there was a significant ($p = 0.001$) difference in overweight and obesity status relative to gender. The prevalence of overweight or obesity among male adolescents (54.3%) was significantly ($p = 0.010$) higher than among females (44.6%). However, there was no significant ($p = 0.731$) difference in the prevalence of obesity relative to age from 15 to 18 years old (not shown in the table); the proportions of overweight plus obesity were 53.3%, 47.5%, 48.3%, and 52.7% for 15 to 18 years, respectively.

**Table 1. Anthropometric characteristics of the participating adolescents relative to gender.**

| Variable | All | Males | Females | p-value [*] |
|---|---|---|---|---|
| | N = 706 | N = 343 | N = 363 | |
| Age | 16.5 ± 0.94 | 16.5 ± 0.89 | 16.5 ± 0.98 | 0.466 |
| Body weight (kg) | 69.2 ± 22.5 | 77.1 ± 24.4 | 61.8 ± 17.6 | < 0.001 |
| Height (cm) | 162.7 ± 9.4 | 169.8 ± 7.1 | 155.9 ± 5.6 | < 0.001 |
| Body mass index (kg/m$^2$) | 26.0 ± 7.4 | 26.6 ± 7.9 | 25.4 ± 6.7 | 0.029 |
| Overweight or obesity status (%) | | | | |
| Underweight | 7.0 | 9.1 | 5.0 | 0.001 |
| Normal weight | 43.7 | 36.5 | 50.4 | |
| Overweight | 24.0 | 25.4 | 22.6 | |
| Obesity | 25.4 | 28.9 | 22.0 | |
| Overweight plus obesity | 49.4 | 54.3 | 44.6 | 0.010 |

Data are means ± standard deviations or percentage.

[*] T-test for independent samples or Chi Squares tests for the proportion.

**Table 2. Two-way ANOVA for BMI among male and female adolescents by age category.**

| Gender | Age Groups (years) | | | | | p-value |
|---|---|---|---|---|---|---|
| | **15** | **16** | **17** | **18** | **All** | |
| **Male** | 25.1 ± 8.4 | 26.0 ± 7.2 | 26.5 ± 7.1 | 29.7 ± 10.5 | 26.6 ± 7.9 | Age: **0.038** |
| | | | | | | Gender: **0.023** |
| **Female** | 25.5 ± 7.4 | 25.2 ± 6.7 | 25.3 ± 6.5 | 25.9 ± 6.6 | 25.4 ± 6.7 | Age by gender interactions: 0.155 |

Table 2 presents the findings of the two-way ANOVA for BMI by gender and age group. There were significant differences in BMI in the main effects of age (p = 0.038) and gender (p = 0.023), but not in the interactions of age with gender (p = 0.155).

The results of the influence of overweight or obesity status on adolescent's perception of how their parents or friends see their weight status are shown in Table 3. There was a significant trend toward a more successful estimation of an adolescent's overweight or obesity by parents or friends. However, there was no significant difference relative to overweight or obesity teasing by either parents or friends. About 37% of adolescents with overweight or obesity were teased very often by parents compared to about 30% without overweight or obesity. Also, 32.4% of adolescents with overweight or obesity were teased by friends. In contrast, 25.6% of adolescents without overweight or obesity were teased by friends.

Table 4 presents the findings of a multivariate analysis of selected variables while controlling for the effect of age and stratified by gender and obesity status. There were significant

**Table 3. Results of cross tabulation of selected variables related to adolescents' perception of how parents' or friends' see their weight status relative to an adolescent's overweight or obesity versus non-overweight/non-obesity status.**

| Variable | Non-overweight or non-obesity | Overweight or obesity | p-value * |
|---|---|---|---|
| **How do your parents classify your weight?** | | | **< 0.001** |
| Underweight | 37.7 | 5.2 | |
| Normal weight | 47.9 | 26.2 | |
| Overweight | 13.0 | 54.7 | |
| Obesity | 1.4 | 14.0 | |
| **Do any of your parents tease you because of your weigh?** | | | 0.152 |
| Very often | 29.9 | 36.8 | |
| Sometimes | 69.5 | 62.6 | |
| Seldom/none | 0.6 | 0.6 | |
| **Do your parents compare your weight with any of your brother/sister?** | | | 0.249 |
| I have no brother or sister | 3.1 | 4.3 | |
| Yes | 36.1 | 40.8 | |
| No | 60.8 | 54.9 | |
| **How do your friends classify your weight?** | | | **< 0.001** |
| Underweight | 38.0 | 5.2 | |
| Normal weight | 57.8 | 33.5 | |
| Overweight | 4.0 | 48.4 | |
| Obesity | 0.3 | 12.8 | |
| **Do any of your friends tease you because of your weigh?** | | | 0.139 |
| Very often | 25.6 | 32.4 | |
| Sometimes | 74.1 | 67.3 | |
| Seldom/none | 0.3 | 0.3 | |

* Chi Squares tests for the differences in proportions between non-overweight or non-obesity and overweight or obesity categories.

**Table 4. Multivariable analysis of selected variables stratified by gender and obesity status while controlling for age.**

| Variable | Gender | Non-overweight/non-obesity | Overweight/obesity | p-value * |
|---|---|---|---|---|
| | | | | Between subjects effects |
| **Body weight** (kg) | **Male** | 58.3 ± 18.7 | 93.6 ± 21.9 | Age: **0.010**; Gender: < **0.001**; Obesity status: 0.839; Gender by obesity interaction: < **0.001** |
| | **Female** | 50.5 ± 7.2 | 74.9 ± 16.6 | |
| | **All** | 53.8 ± 8.7 | 84.9 ± 21.7 | |
| **Height** (cm) | **Male** | 168.0 ± 7.4 | 171.1 ± 7.0 | Age: 0.248; Gender: < **0.001**; Obesity status: **0.003**; Gender by obesity interaction: 0.079 |
| | **Female** | 155.7 ± 5.4 | 156.2 ± 5.9 | |
| | **All** | 161.1 ± 9.0 | 164.2 ± 9.6 | |
| **BMI** (kg/m2) | **Male** | 20.4 ± 2.4 | 31.9 ± 7.3 | Age: **0.011**; Gender: 0.391; Obesity status: < **0.001**; Gender by obesity interaction: **0.042** |
| | **Female** | 20.9 ± 2.2 | 30.7 ± 6.3 | |
| | **All** | 20.7 ± 2.3 | 31.4 ± 6.9 | |
| **Age at puberty** (years) | **Male** | 13.4 ± 1.2 | 13.3 ± 1.2 | Age: < **0.001**; Gender: <0.001; Obesity status: 0.062; Gender by obesity interaction: 0.696 |
| | **Female** | 12.7 ± 1.3 | 12.5 ± 1.2 | |
| | **All** | 13.0 ± 1.3 | 12.9 ± 1.2 | |

Data are means and standard deviations.

* Wilks' Lambda p values for the main effects of all of the variables are < 0.001; gender by obesity status interactions < 0.001.

gender by obesity interactions in body weight (p < 0.001) and BMI (p = 0.042) but not in age at puberty (p-value for the interaction = 0.696). However, age at puberty showed a significant main effect for age with a p-value < 0.001.

Finally, logistic regression analysis, adjusted for age and gender, for selected variables relative to non-overweight/non-obesity or overweight/obesity is presented in Table 5. There was a significant gender difference relative to overweight/obesity status, as females are less likely to be overweight or obese (aOR = 0.508; 95% CI = 0.327–0.789; p = 0.003). Compared to normal weight, adolescents perceived their parents as more likely to classify their weight as overweight (aOR = 4.417; 95% CI = 1.404–13.895; p = 0.011) or obese (aOR = 9.950; 95% CI = 2.847–34.766; p < 0.001). Adolescents were also more likely to see their friends classify the weight as overweight (aOR = 26.292; 95% CI = 3.336–207.192; p = 0.002) or obese (aOR = 81.850; 95% CI = 9.783–684.791; p < 0.001).

**Table 5. Logistic regression analysis, adjusted for age and gender, for selected variables relative to non-overweight/non-obesity and overweight/obesity.**

| Variable | Non-overweight/non-obesity versus Overweight/obesity * | | | |
|---|---|---|---|---|
| | aOR | (95% CI) | SEE | p-value |
| **Age** | 0.873 | 0.694–1.097 | 0.117 | 0.244 |
| **Gender** (boys = ref) | 1.00 | | | |
| Girls | 0.508 | 0.327–0.789 | 0.225 | **0.003** |
| **How do your parents classify your weight?** (underweight = ref) | 1.00 | | | |
| Normal weight | 1.348 | 0.416–4.371 | 0.600 | 0.619 |
| Overweight | 4.417 | 1.404–13.895 | 0.585 | **0.011** |
| Obesity | 9.950 | 2.847–34.766 | 0.638 | < **0.001** |
| **How do your friends classify your weight?** (underweight = ref) | 1.00 | | | |
| Normal weight | 2.516 | 0.299–21.158 | 1.086 | 0.396 |
| Overweight | 26.292 | 3.336–207.192 | 1.053 | **0.002** |
| Obesity | 81.850 | 9.783–684.791 | 1.084 | < **0.001** |

* Non-overweight/non-obese was used as a reference category. aOR = adjusted odds ratio; CI = confidence interval; ref = reference category; SEE = standard error.

## Discussion

The present study intended to provide an update on the rate of overweight or obesity among Kuwaiti adolescents and examine the relationship of measured overweight or obesity status with adolescents' perception of how their parents or friends see their weight status. The main findings indicated that the prevalence of overweight or obesity among Kuwaiti adolescents was nearly 50%, with males having significantly higher overweight or obesity than females. However, there was no significant difference in the prevalence of obesity relative to age from 15 to 18 years old. Also, the findings showed that there was a significant trend toward a more successful estimation of an adolescents' perception of their overweight or obesity as seen by parents or friends. In addition, logistic regression analysis, adjusted for age and gender, revealed significant differences relative to overweight/obesity status, as females were 50% less likely to be overweight or obese. Further, adolescents perceived their parents or friends as more likely to classify their weight as overweight or obese.

Our findings on the rate of overweight and obesity in Kuwaiti adolescents are in agreement with those from other studies published over the past 15 years [2–4]. This could indicate that adolescent overweight or obesity has reached a plateau in recent years in the State of Kuwait. A comparison of our findings with those of the regional prevalence of overweight or obesity also showed similar results [24–28]. However, a recent study in Oman reported much lower overweight/obesity levels for Omani adolescent females (22.4%) and males (22.5%) [29]. The high prevalence of overweight or obesity in adolescents in the State of Kuwait and neighboring Gulf countries highlights the importance of improving lifestyle behaviors, including nutritional and exercise education in this region. Previous studies indicated that reasons for increased obesity levels among adolescents in the Gulf countries were linked to high fast food consumption, low physical activity levels, high sedentary behaviors, and increased intake of high-fat foods [30–32].

The present findings found a higher prevalence of overweight or obesity in adolescent males compared to females. These findings are consistent with the results of a cross-sectional survey conducted in the United Arab Emirates (UAE) [33]. Also, a recent systematic review revealed a very high prevalence of obesity among children and adolescents in the Gulf Cooperation Council (GCC) countries, and that the prevalence increased with age, and was consistently higher in boys than girls [34]. In the current study, however, there was no significant difference in the prevalence of obesity relative to age from 15 to 18 years old. Globally, the gender difference in the prevalence of obesity among adolescents varies by country and region. BMI centiles among 15-year old adolescents in an international sample showed statistically higher rates among males in some countries (United States: 13.9%, and Greek: 10.8%) and higher prevalence among females in other countries (United States: 15.1%, and Portugal: 6.7%) [35]. Another reason for the gender differences in overweight or obesity prevalence is that the BMI cut-offs used for children and adolescents are not standardized across studies. This makes it difficult to compare obesity rates between studies because of the impact on the proportion of children defined as overweight or obese [36]. A systematic review has concluded that the gender differences in obesity rates as defined by BMI in childhood and adolescence can be traced to differences in biology, society, and culture [37].

The current study indicated that there was a significant trend toward more successful estimation of an adolescent's overweight or obesity by parents or friends, as logistic regression analysis, adjusted for age and gender, showed that parents and friends were significantly more likely to classify adolescents with overweight/obesity as overweight or obese. An Italian study involving children with a diverse range of body weights and obesity found that almost half of parents classified the adolescents' weight status incorrectly and that the probability of parents correctly estimating weight status decreased with increasing BMI percentiles and was lower

with boys than girls [38]. Parental underestimation of child weight status was also reported in a study from Texas, USA, where it seemed to be common among a cross-sectional sample of children and parents; the study recommended improving the parental perception of child weight [39]. Important moderators of parental underestimation of children's overweight or obesity status were found to include parent weight, child gender, and the assessment method used (visual versus nonvisual) [40]. Furthermore, identified obesity-associated factors among Korean male and female students included underestimation and overestimation of male adolescent weight by fathers, whereas factors among females included underestimation of female adolescent weight by fathers [41]. The study concluded that parental involvement in obesity interventions could aid in preventing adolescent obesity [41].

Although there were no significant differences found in the present study in teasing adolescents because of overweight or obesity status among parents or friends, overweight or obese adolescents reported being teased "very often" due to their weight by parents (36.8%) or friends (32.4%), which was considered high. Based on the findings of a recent study, weight-related teasing is one of the most common forms of bullying in schools, particularly for obese adolescents [42]. According to a qualitative analysis by Berge et al., adolescents experienced weight-related teasing in two-thirds of households [43]. Such findings may suggest that household joking or teasing has a negative impact on an adolescent's body appearance and self-esteem. Negative body image talk has been related to increased rates of overweight and obesity among adolescents, and therefore efforts to educate parents about the negative impact of weight-related teasing are necessary [44, 45]. Also, a longitudinal study found that weight-related teasing was quite prevalent among adolescents with overweight or obesity and that teasing from female peers led to a stronger negative relationship with weight esteem for youth with average weight [46]. In a qualitative study, some of the threats and concerns expressed by obese adolescents included lack of support from parents, trusted friends, and health-care providers and bullying, shame, guilt, and self-blame [47]. However, obese adolescents were found to perceive peers as a possible source of support [47].

The strengths of the current study included having a representative and adequate sample size. The study also used a previously validated questionnaire. Further, the study examined how parents (or friends) perceived adolescents' weight status as seen by the adolescents themselves; other studies looked to direct perception of parents or friends to adolescent's weight status [38–40, 43]. However, this study did have some limitations. Access to socio-economic data would have provided insight into the influences of the adolescents' background, parents' education, and household income. Self-reported behavioral data may also introduce bias, with the potential for under- or overestimation. In addition, the study was cross-sectional, which precludes us from inferring a causal relationship between the selected variables. In addition, physical activity was not assessed, a variable that can influence overweight or obesity status. It should also be noted that perceptions may be different in this age group (adolescents) toward themselves and others, as teenagers tend to be more sensitive at this age toward body image teasing.

## Conclusion

Overweight or obesity levels appear high among Kuwaiti adolescents; about 50% of adolescents are overweight or obese, with males having significantly higher overweight or obesity than females. Also, there was a significant trend toward a more successful estimation of an adolescents' perception of their overweight or obesity as seen by parents or friends. In addition, logistic regression analysis, adjusted for age and gender, revealed significant differences relative to overweight/obesity status, as females were 50% less likely to be overweight or obese. Further,

adolescents perceived their parents or friends as more likely to classify their weight as overweight or obese. The present study updated the information related to the prevalence of overweight or obesity among Kuwaiti adolescents and showed the importance of parent's and friend's perception of adolescent's weight status.

## Supporting information

**S1 Raw data.**
(XLS)

## Acknowledgments

We would like to thank the Ministry of Education in Kuwait for their approval to conduct the study in schools and to all the participating students for taking part in this study.

## Author Contributions

**Conceptualization:** Ahmad R. Al-Haifi.

**Data curation:** Ahmad R. Al-Haifi, Balqees A. Al-Awadhi, Yousef A. Al-Dashti, Badriyah H. Aljazzaf, Ahmad R. Allafi.

**Formal analysis:** Hazzaa M. Al-Hazzaa.

**Investigation:** Ahmad R. Al-Haifi, Balqees A. Al-Awadhi, Yousef A. Al-Dashti, Badriyah H. Aljazzaf, Ahmad R. Allafi, Mariam A. Al-Mannai, Hazzaa M. Al-Hazzaa.

**Methodology:** Ahmad R. Al-Haifi, Balqees A. Al-Awadhi, Yousef A. Al-Dashti, Badriyah H. Aljazzaf, Ahmad R. Allafi, Mariam A. Al-Mannai, Hazzaa M. Al-Hazzaa.

**Project administration:** Ahmad R. Al-Haifi.

**Resources:** Ahmad R. Allafi.

**Supervision:** Ahmad R. Al-Haifi, Balqees A. Al-Awadhi, Yousef A. Al-Dashti, Badriyah H. Aljazzaf, Ahmad R. Allafi.

**Writing – original draft:** Ahmad R. Al-Haifi.

**Writing – review & editing:** Balqees A. Al-Awadhi, Yousef A. Al-Dashti, Badriyah H. Aljazzaf, Ahmad R. Allafi, Mariam A. Al-Mannai, Hazzaa M. Al-Hazzaa.

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
