## [Decision Letter · Decision Letter 0]

12 Aug 2021

PONE-D-21-14400

Prevalence of Obesity among Kuwaiti Adolescents and its Relationship with their Perception of Body Weight as seen by Parents or Friends

PLOS ONE

Dear Dr. Al-Hazzaa,

Thank you for submitting your manuscript to PLOS ONE. After careful consideration, we feel that it has merit but does not fully meet PLOS ONE’s publication criteria as it currently stands. Therefore, we invite you to submit a revised version of the manuscript that addresses the points raised during the review process.

We look forward to receiving your revised manuscript.

Kind regards,

Shahrad Taheri

Academic Editor

PLOS ONE

1. Please ensure that your manuscript meets PLOS ONE's style requirements, including those for file naming. The PLOS ONE style templates can be found at https://journals.plos.org/plosone/s/file?id=wjVg/PLOSOne_formatting_sample_main_body.pdf and https://journals.plos.org/plosone/s/file?id=ba62/PLOSOne_formatting_sample_title_authors_affiliations.pdf.

3. Thank you for stating the following in the Acknowledgments/Funding Section of your manuscript:

“Professor Hazzaa M. Al-Hazzaa’s research has been funded by the Deanship of Scientific Research at Princess Nourah bint Abdulrahman University through the Fast-track Research Funding Program.”

Additional Editor Comments (if provided):

Reviewers' comments:

Reviewer's Responses to Questions

**Comments to the Author**

1. Is the manuscript technically sound, and do the data support the conclusions?

Reviewer #1: Partly

2. Has the statistical analysis been performed appropriately and rigorously? 

Reviewer #1: Yes

3. Have the authors made all data underlying the findings in their manuscript fully available?

Reviewer #1: Yes

4. Is the manuscript presented in an intelligible fashion and written in standard English?

Reviewer #1: Yes

5. Review Comments to the Author

Reviewer #1: This study is interesting but have few points to consider:

Title

I suggest that the title be Prevalence of overweight and obesity among kuwaiti adolescents and the perception of body weight by parents or friends.

Keywords

I suggest replacing “weight estimation” and “obesity perception” with “body weight” and “weight perception”.

Introduction

I suggest reversing the order of references 10 and 11 in the text (lines 85-90).

Present the abbreviation of the word “body mass index” (lines 83-84) instead of putting it into methods (line 132).

Materials and Methods

Study design and participants

The researchers reported that they recruited “adolescents between the ages of 15 and 18 years” (lines 114-115). What was the criterion adopted for choosing this age group?

What was the estimated sample size? Were there losses?

How did the researchers assess whether the students suffered or not from diet-related disorders? (line 122).

Description of questionnaire

The researchers did not described about the questions “Do your parents compare your

weight with any of your brother/sister?” and “age at maturation”. How did you assess the maturation age?

Statistical analysis

In “Descriptive statistics were obtained for all variables...”, include the variables (line 157).

In the sentence “Chi-square tests of proportions were used to test differences in selected variables related to the influence of media on adolescents’ decision to lose weight...”, explain this relation of media influence (lines 160-162).

Results

The Table 3 refers to the weight status from individuals with “non-overweight or non-obesity” or “overweight or obesity”, thus why the title is written “relative to an adolescent’s overweight or obesity”?

Please, correct the 95% CI referring to the data from overweight individuals (line 201).

Discussion

The researchers mentioned that “...higher rates among males in some countries and higher prevalence among females in other countries...”, I think it is interesting to include these countries (lines 238-241).

The sentence mention about female adolescents and, what about female students? (lines 260-263).

Why did not the researchers assess the socioeconomic conditions, physical activity and food consumption? I think it is valid that you reassess and use these aspects as variables, due they are very important and can influence in the nutritional status of adolescents and in the weight perception.

Include the contributions that this study give to the society.

6. PLOS authors have the option to publish the peer review history of their article (what does this mean?). If published, this will include your full peer review and any attached files.

Reviewer #1: No

---

## [Author Response · Author response to Decision Letter 0]

15 Aug 2021

All responses are provided in the file titled "response to te reviewers' comments".

---

## [Decision Letter · Decision Letter 1]

6 Oct 2021

PONE-D-21-14400R1Prevalence of overweight and obesity among Kuwaiti adolescents and the perception of body weight by parents or friendsPLOS ONE

Dear Dr. Al-Hazzaa,

Thank you for submitting your manuscript to PLOS ONE. After careful consideration, we feel that it has merit but does not fully meet PLOS ONE’s publication criteria as it currently stands. Therefore, we invite you to submit a revised version of the manuscript that addresses the points raised during the review process.

We look forward to receiving your revised manuscript.

Kind regards,

Shahrad Taheri

Academic Editor

PLOS ONE

Journal Requirements:

Additional Editor Comments (if provided):

Reviewers' comments:

Reviewer's Responses to Questions

**Comments to the Author**

1. If the authors have adequately addressed your comments raised in a previous round of review and you feel that this manuscript is now acceptable for publication, you may indicate that here to bypass the “Comments to the Author” section, enter your conflict of interest statement in the “Confidential to Editor” section, and submit your "Accept" recommendation.

Reviewer #1: (No Response)

2. Is the manuscript technically sound, and do the data support the conclusions?

Reviewer #1: Yes

3. Has the statistical analysis been performed appropriately and rigorously? 

Reviewer #1: Yes

4. Have the authors made all data underlying the findings in their manuscript fully available?

Reviewer #1: (No Response)

5. Is the manuscript presented in an intelligible fashion and written in standard English?

Reviewer #1: No

6. Review Comments to the Author

Reviewer #1: ABSTRACT

-The values in this sentence “...perceived their parents (p < 0.001) or friends (p < 0.001) as more likely to classify their weight as overweight or obese” (lines 52-53) refer to the obese individuals, not the overweight people (parents: p = 0.011 and friends: p = 0.002).

MATERIALS AND METHODS

-Describe in the manuscript when the data collection was performed.

-“Chi-square tests of independence was used to examine the relationships between overweight or obesity category and selected variables related to the influence of media on adolescents’ decision to lose weight or their perception of body thinness” (lines 164-166).

It is not clear the sentence above on the use of variables related to the influence of media on adolescents’ decision to lose weight or their perception of body thinness, because the researchers didn’t show the results/discussed about it. Furthermore, in lines 157-158 it is described that “In the present study, we used the data from part three only, which was related to body weight perception by parents or friends” and it is the part two that is related to “influence of media on dieting to lose weight”. Explain.

RESULTS

-The 95% CI datum cited in the line 208 about overweight individuals is not compatible with the informed in the Table 5.

DISCUSSION

-In the sentence “other studies published over the past ‘seven’ years [2-4]” (line 226) the reference of number 4 was published in 2004. Change the description of years.

-The researchers assessed the adolescents’ perceptions of parents’ and peers’ opinions about their weight. I think it is valid to add other studies that have assessed the same (the adolescents’ perceptions of how parents and peers assess their weight). And it would be interesting to describe the reason whereby you chose to collect this information from the adolescent, not by parents/peers. Also, in the conclusion section the researchers mentioned of importance of parents’ perception (and why not from friends?), lines 311-312.

TABLE 1 - Wouldn’t the variable “overweight or obesity status (%)” be “nutritional status”?

Notes:

-Please, when checking the mentioned lines use the manuscript with tracked changes.

-Please, review the manuscript writing, words as “Portuguese” (line 248), sentences (as in lines 270-273), among others.

7. PLOS authors have the option to publish the peer review history of their article (what does this mean?). If published, this will include your full peer review and any attached files.

Reviewer #1: No

---

## [Author Response · Author response to Decision Letter 1]

18 Oct 2021

Responses to the Reviewers Comments- Round -2

1- Reviewer #1: ABSTRACT

-The values in this sentence “...perceived their parents (p < 0.001) or friends (p < 0.001) as more likely to classify their weight as overweight or obese” (lines 52-53) refer to the obese individuals, not the overweight people (parents: p = 0.011 and friends: p = 0.002). We modified the sentence to be as follow:

Authors’ Response

We modified the sentence to be as follow:

“… that adolescents perceived their parents (p = 0.011 and p < 0.001) or friends (p = 0.002 and p < 0.001) as more likely to classify their weight as overweight or obese, respectively.

2- MATERIALS AND METHODS

-Describe in the manuscript when the data collection was performed. 

Authors’ Response

The data collection was performed during the fall of 2019.

3- “Chi-square tests of independence was used to examine the relationships between overweight or obesity category and selected variables related to the influence of media on adolescents’ decision to lose weight or their perception of body thinness” (lines 164-166). It is not clear the sentence above on the use of variables related to the influence of media on adolescents’ decision to lose weight or their perception of body thinness, because the researchers didn’t show the results/discussed about it. 

Authors’ Response

The statement was modified to the following:

Chi-square test of independence was used to examine the relationships of selected variables related to adolescents’ perception of how parents’ or friends’ see their weight status relative to an adolescent’s overweight or obesity versus non-overweight/non-obesity status. 

The results are shown in table 3. 

4- Furthermore, in lines 157-158 it is described that “In the present study, we used the data from part three only, which was related to body weight perception by parents or friends” and it is the part two that is related to “influence of media on dieting to lose weight”. Explain. 

Authors’ Response

It is correct that the data were from part three. However, we slightly modified the sentences to read as follow:

In the present study, we used the data from part three only, which was related to adolescent’s body weight perception as seen by parents or friends.

5- RESULTS

-The 95% CI datum cited in the line 208 about overweight individuals is not compatible with the informed in the Table 5. 

Authors’ Response

Thanks for the comments. We have corrected the wrong 95% CI datum shown in page 7, line 202.

6- DISCUSSION

-In the sentence “other studies published over the past ‘seven’ years [2-4]” (line 226) the reference of number 4 was published in 2004. Change the description of years. 

Authors’ Response

We corrected the sentence so to read as follow:

 “… other studies published over the past 15 years [2-4].”

7- -The researchers assessed the adolescents’ perceptions of parents’ and peers’ opinions about their weight. I think it is valid to add other studies that have assessed the same (the adolescents’ perceptions of how parents and peers assess their weight). And it would be interesting to describe the reason whereby you chose to collect this information from the adolescent, not by parents/peers. 

Authors’ Response

Using PubMed and Google scholar search, we were unable to locate any study related to how parents (or friends) perceived adolescents’ weight status as seen by the adolescents themselves. This is the major reason lead to the initiation of this study. It is also more important to assess the direct perception of the adolescents not the parents (or friend) about how they see other perception about their weight status. 

We added a sentence in the strength of the present study indicating this aspect of the study.

8- Also, in the conclusion section the researchers mentioned of importance of parents’ perception (and why not from friends?), lines 311-312. 

Authors’ Response

Thanks. We added friends as well. 

9- TABLE 1 - Wouldn’t the variable “overweight or obesity status (%)” be “nutritional status”? 

Authors’ Response

It can be both titles. However, we have chosen “overweight or obesity” as this is the stated definition in the extended International Obesity Task Force (IOTF) age- and sex-specific BMI cutoff reference standards [reference number 22].

10- Please, review the manuscript writing, words as “Portuguese” (line 248), sentences (as in lines 270-273), among others. 

Authors’ Response

Portuguese was corrected to Portugal.

---

## [Editor Report · Decision Letter 2]

17 Dec 2021

Prevalence of overweight and obesity among Kuwaiti adolescents and the perception of body weight by parents or friends

PONE-D-21-14400R2

Dear Dr. Al-Hazzaa,

We’re pleased to inform you that your manuscript has been judged scientifically suitable for publication and will be formally accepted for publication once it meets all outstanding technical requirements.

Kind regards,

Shahrad Taheri

Academic Editor

PLOS ONE
---

## [Editor Report · Acceptance letter]

23 Dec 2021

PONE-D-21-14400R2 

Prevalence of overweight and obesity among Kuwaiti adolescents and the perception of body weight by parents or friends 

Dear Dr. Al-Hazzaa:

I'm pleased to inform you that your manuscript has been deemed suitable for publication in PLOS ONE. Congratulations! Your manuscript is now with our production department. 

Kind regards, 

on behalf of

Dr. Shahrad Taheri 

Academic Editor

PLOS ONE